# NADPH oxidase mediates microtubule alterations and diaphragm dysfunction in dystrophic mice

James Anthony Loehr[1], Shang Wang[1], Tanya R Cully[1], Rituraj Pal[1], Irina V Larina[1], Kirill V Larin[1,2,3], George G Rodney[1]*

[1]Department of Molecular Physiology and Biophysics, Baylor College of Medicine, Houston, United States; [2]Department of Biomedical Engineering, University of Houston, Houston, United States; [3]Interdisciplinary Laboratory of Biophotonics, Tomsk State University, Tomsk, Russia

**Abstract** Skeletal muscle from *mdx* mice is characterized by increased Nox2 ROS, altered microtubule network, increased muscle stiffness, and decreased muscle/respiratory function. While microtubule de-tyrosination has been suggested to increase stiffness and Nox2 ROS production in isolated single myofibers, its role in altering tissue stiffness and muscle function has not been established. Because Nox2 ROS production is upregulated prior to microtubule network alterations and ROS affect microtubule formation, we investigated the role of Nox2 ROS in diaphragm tissue microtubule organization, stiffness and muscle/respiratory function. Eliminating Nox2 ROS prevents microtubule disorganization and reduces fibrosis and muscle stiffness in *mdx* diaphragm. Fibrosis accounts for the majority of variance in diaphragm stiffness and decreased function, implicating altered extracellular matrix and not microtubule de-tyrosination as a modulator of diaphragm tissue function. Ultimately, inhibiting Nox2 ROS production increased force and respiratory function in dystrophic diaphragm, establishing Nox2 as a potential therapeutic target in Duchenne muscular dystrophy.

*For correspondence:
rodney@bcm.edu

Competing interests: The authors declare that no competing interests exist.

## Introduction

Duchenne muscular dystrophy (DMD) is an X-linked recessive disease which affects 1 in every 3500 boys resulting in progressive muscle atrophy, loss of ambulation and cardio-respiratory failure (*Levi et al., 2015*). In DMD patients, the leading cause of mortality is diaphragm dysfunction (*Finder et al., 2004*; *Finsterer and Stöllberger, 2003*; *Percival et al., 2012*). In the *mdx* animal, a mouse model of DMD, disease progression in the diaphragm mimics the human development of the disease (*Stedman et al., 1991*), and respiratory dysfunction has been shown to promote cardiac dysfunction (*Barbin et al., 2016*; *Finder et al., 2004*; *Lanza et al., 2001*).

NADPH Oxidase 2 (Nox2) has been shown to play an important role in dystrophic muscle. Nox2 content and activity are upregulated prior to the onset of inflammation and necrosis (*Whitehead et al., 2010*) and downregulating Nox2 ROS production protects against pathophysiological alterations in young (5–7 wk) dystrophic muscle (*Pal et al., 2014*). Recent evidence indicates the microtubule (MT) network is dysregulated in dystrophic muscle (*Belanto et al., 2016*; *Iyer et al., 2017*; *Khairallah et al., 2012*; *Prins et al., 2009*), which results in aberrant Nox2 ROS production and implicates Nox2 ROS in altered mechanotransduction (*Khairallah et al., 2012*). However, Nox2 ROS is upregulated early (19 d; (*Whitehead et al., 2010*)), prior to changes in the MT network (*Belanto et al., 2016*; *Iyer et al., 2017*; *Khairallah et al., 2012*; *Prins et al., 2009*), and oxidation has been shown to be a post-translational modification of the MT network (*Clark et al., 2014*;

*Landino et al., 2004*; *Wilson and González-Billault, 2015*). These findings raise the question of whether Nox2 ROS initiates changes in the MT network.

In addition to increased Nox2 ROS production and alterations in the MT network, dystrophic muscle is characterized by increased fibrosis and muscle stiffness (*Cornu et al., 1998*; *Cornu et al., 2001*; *Virgilio et al., 2015*). The de-tyrosination of α-tubulin (DT-tubulin) has been proposed as a mechanism which prevents the de-polymerization of the MT network, causing an increase in muscle stiffness and dysfunction in isolated muscle cells (*Kerr et al., 2015*; *Robison et al., 2016*). However, Ervasti and colleagues (*Belanto et al., 2016*) demonstrated increased muscle stiffness with no differences in relative DT-tubulin amounts between *mdx* and WT mice. MT formation is also sensitive to the extracellular environment (*Myers et al., 2011*; *Putnam et al., 2003*; *Putnam et al., 2001*) and increased extracellular matrix (ECM) has been implicated in increased muscle stiffness and decreased force production (*Desguerre et al., 2009*; *Meyer and Lieber, 2011*; *Percival et al., 2012*; *Rowe et al., 2010*; *Wood et al., 2014*). Intriguingly, transgenic *mdx* mice expressing either a nearly full length dystrophin (Dys$^{\Delta71-78}$-*mdx*) or overexpressing utrophin (*Fiona*) suggest that MT density and organization is independent of the level of MT de-tyrosination (*Belanto et al., 2014*; *Belanto et al., 2016*). Taken together, the role of de-tyrosinated MTs in tissue stiffness and disease pathogenesis in muscular dystrophy is unclear.

Skeletal muscle stiffness traditionally has been evaluated using either atomic force microscopy (AFM; [*Canato et al., 2010*; *Kerr et al., 2015*; *Mathur et al., 2001*; *van Zwieten et al., 2014*]) or the passive properties of muscle measured during stretch (*Hakim and Duan, 2013*; *Hakim et al., 2011*; *Lopez et al., 2008*; *Rowe et al., 2010*). AFM evaluates single muscle fiber stiffness but does not consider cell-cell interactions or the influence of the extra cellular matrix. While evaluating stiffness through muscle passive properties considers the series and parallel elastic components together it does not differentiate between the contributions of longitudinal (series) or transverse (parallel) tissue stiffness within overall muscle stiffness. Optical coherence elastography (OCE) recently has been developed as a unique method to noninvasively evaluate tissue stiffness (*Larin and Sampson, 2017*; *Wang and Larin, 2014*; *Wang et al., 2012*; *Wang et al., 2014*). Here, we utilize OCE to evaluate the differences in longitudinal and transverse tissue stiffness in the diaphragm of *mdx* mice. Previous data indicate *mdx* muscle is compromised in the transverse direction (*Kumar et al., 2004*; *Ramaswamy et al., 2011*). Therefore, OCE may provide a unique method to differentiate pathological alterations in longitudinal and transverse stiffness and their impact on muscle function.

Because the altered MT network and fibrosis develop later in the disease pathology, after Nox2 ROS production has been initiated, we hypothesized that genetically eliminating Nox2 ROS production would prevent alterations to the MT network and reduce diaphragm stiffness thereby improving muscle and respiratory function in adult *mdx* mice. We also hypothesized, at the tissue level, stiffness would be greater in the transverse direction and fibrosis would be the major determinant of tissue stiffness.

## Results

### Genetic deletion of Nox2 ROS production prevents disorganization of the microtubule network in dystrophic muscle

Previous data have shown that tubulin content is upregulated in muscular dystrophy, and DT-tubulin may influence MT stability (*Kerr et al., 2015*; *Khairallah et al., 2012*; *Prins et al., 2009*). However, *Belanto et al. (2016)* have suggested that the relative DT-tubulin level is not elevated in *mdx* muscle. Our data confirm that α-, β-, and DT-tubulin are elevated with muscular dystrophy and extend these findings to show that eliminating Nox2 ROS production in *mdx* mice prevents the increase in all three forms of tubulin (*Figure 1B–D*). Because DT-tubulin is the de-tyrosinated form of α-tubulin, and both DT- and α-tubulin are elevated in *mdx* muscle, we assessed the fraction of α-tubulin that is de-tyrosinated. We found that there is no difference in the DT-/α-tubulin ratio between groups (*Figure 1E*), suggesting that the increase in DT-tubulin is likely due to increased α-tubulin. *Khairallah et al. (2012)* demonstrated Nox2 ROS production is increased in response to a polymerized MT network. We found that Nox2 ROS production leads to increased MT disorganization (*Figure 1G–H*) and density (*Figure 1I*) in dystrophic diaphragm muscle which was prevented by

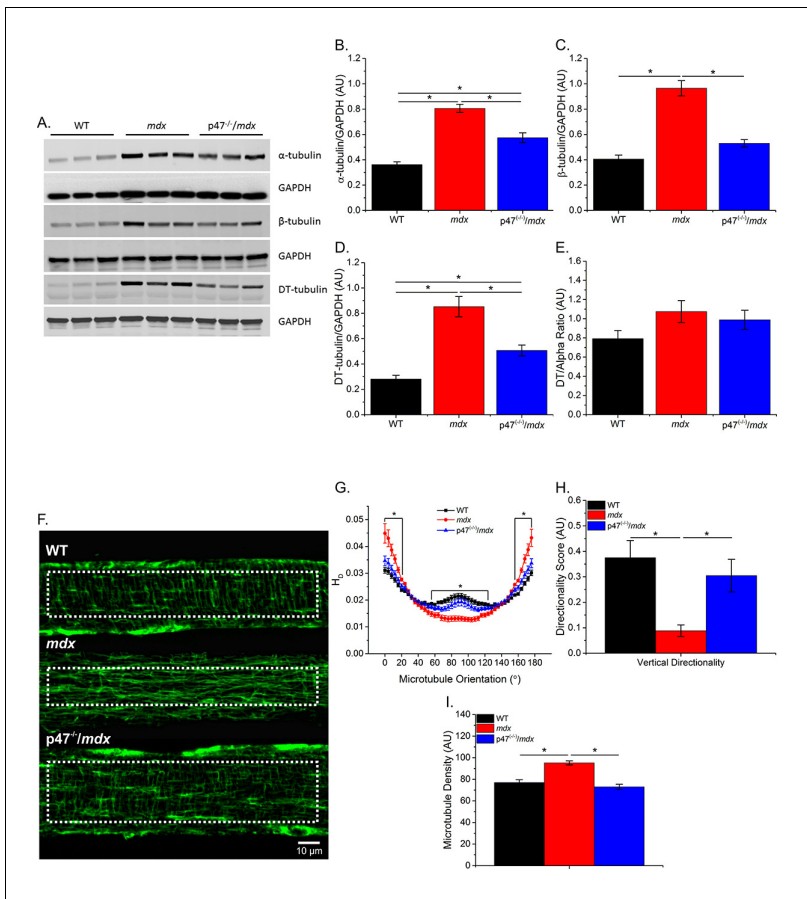

**Figure 1.** Eliminating Nox 2 ROS production prevents alterations in tubulin content and the microtubule network. (A) Representative western blot images of α-, β-, and DT-tubulin content in all three genotypes. (B–D) Eliminating Nox2 ROS production decreases absolute α-, β- and DT-tubulin content in dystrophic diaphragm muscle. (E) The relative amount of DT-/α-tubulin is not different between groups. (F) Representative images of diaphragm myofibers stained with α-tubulin. (G–I) The lack of Nox 2 ROS prevents microtubule disorganization and the increase in microtubule density seen in *mdx* muscle. $p \leq 0.05$ *Significant difference between groups in at least (A–E) $n_{animals} = 6$ and (F–I) $n_{animals} = 3$ and $n_{fibers} = 15$.

eliminating Nox2 ROS. These results indicate that Nox2-generated ROS increases tubulin content, MT disorganization and MT polymerization in dystrophic diaphragm muscle and questions the role of DT-tubulin in MT stabilization or density.

## Genetic inhibition of Nox2 ROS decreases skeletal muscle fibrosis

Increased fibrosis is a pathological hallmark of muscular dystrophy. In accordance with previous studies, we observed increased diaphragm fibrosis in *mdx* compared with WT mice (*Figure 2*). Eliminating Nox2 ROS in dystrophic muscle resulted in reduced collagen as measured by Trichrome staining (*Figure 2A*), hydroxyproline concentration(*Figure 2B*), and collagen I content (*Figure 2C*) as well as fibronectin content (*Figure 2C*). These data suggest that decreasing Nox2 ROS results in a significant decrease in fibrosis in the *mdx* diaphragm.

## Muscle stiffness and stretch induced ROS are reduced in Nox2 deficient dystrophic muscle

Microtubules have been shown to be sensitive to the extracellular environment (*Myers et al., 2011*; *Putnam et al., 2003*; *Putnam et al., 2001*) and cell-to-cell (transverse) interactions are critical in skeletal muscle force transduction (*Passerieux et al., 2007*; *Purslow and Trotter, 1994*; *Ramaswamy et al., 2011*). We evaluated the role of Nox2 ROS in diaphragm mechanical properties

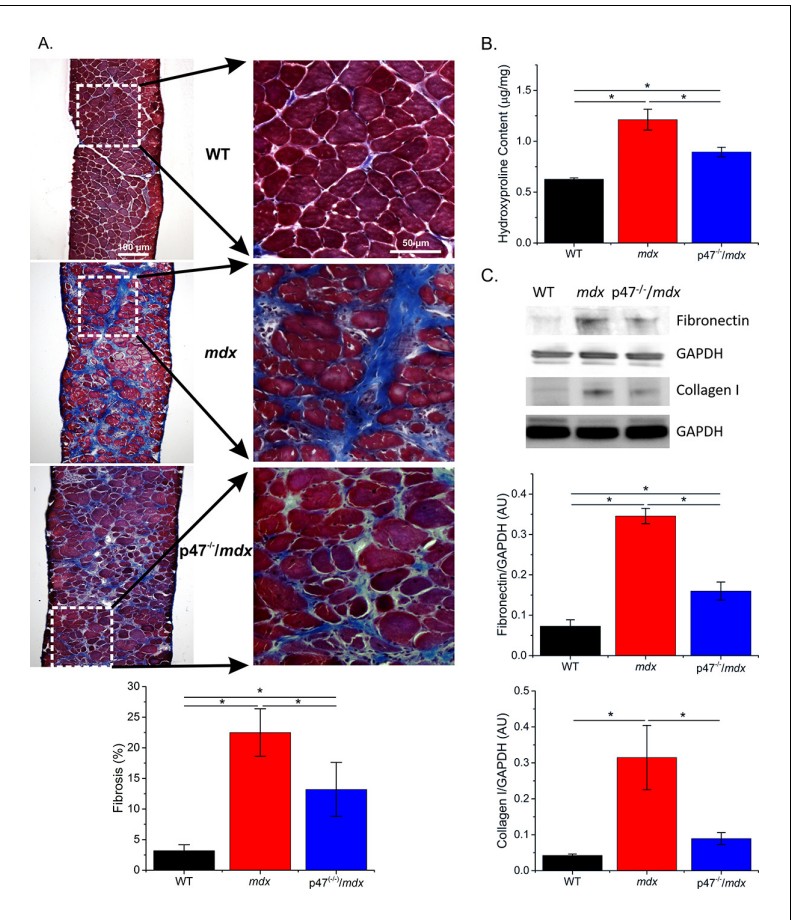

**Figure 2.** Genetic deletion of Nox2 ROS production reduced fibrosis. (**A**) Representative trichrome images of fibrosis in all three genotypes. Eliminating Nox2 ROS production in dystrophic muscle reduced fibrosis compared with *mdx* mice. (**B**) Hydroxyproline levels were elevated in dystrophic muscle and eliminating Nox2 ROS reduced hydroxyproline content compared with *mdx* mice. (**C**) Representative western blot images for fibronectin and collagen I content in all three genotypes. Fibronectin and collagen I content were elevated in *mdx* diaphragm and eliminating Nox2 ROS reduced both toward WT levels. $p \leq 0.05$ * Significant difference between groups in at least $n_{animals} = 6$ for trichrome and hydroxyproline and $n_{animals} = 3$ for fibronectin and collagen I.

using two distinct methods: passive stretch to evaluate the series and parallel elastic components together and optical coherence elastography (OCE) to differentiate between the contributions of series (longitudinal stiffness) and parallel (transverse stiffness) components within overall muscle tissue stiffness. *Figure 3A and E* demonstrate the system design for both passive stretch and OCE, respectively, *Figure 3F* shows a sample OCT image of the diaphragm and *Figure 3—video 1* illustrates a sample wave propagation taken during OCE. Passive stiffness while lengthening the diaphragm to 120% $L_o$ was increased in *mdx* compared with WT mice, and eliminating Nox2 ROS resulted in reduced tissue stiffness compared with *mdx* diaphragm (*Figure 3B–C*). Transverse and longitudinal stiffness, using OCE, was increased in diaphragm of *mdx* mice compared with WT mice. Interestingly, eliminating Nox2 ROS production reduced only longitudinal stiffness in *Ncf1$^{-/-}$::mdx* (designated as p47$^{(-/-)}$/mdx) mice to WT levels (*Figure 3G–H*). Muscle function was measured pre- and post-OCE to ensure OCE measurements did not compromise tissue health. Muscle function for all genotypes was not altered following OCE measurements (*Figure 3I*). We also found that stretch induced ROS was elevated in *mdx* diaphragm compared with both WT and p47$^{(-/-)}$/mdx diaphragm tissue (*Figure 3D*). These data suggest that elevated Nox2 ROS increases diaphragm stiffness in dystrophic muscle and demonstrate Nox2 as the source of stretch induced ROS at the tissue level. In

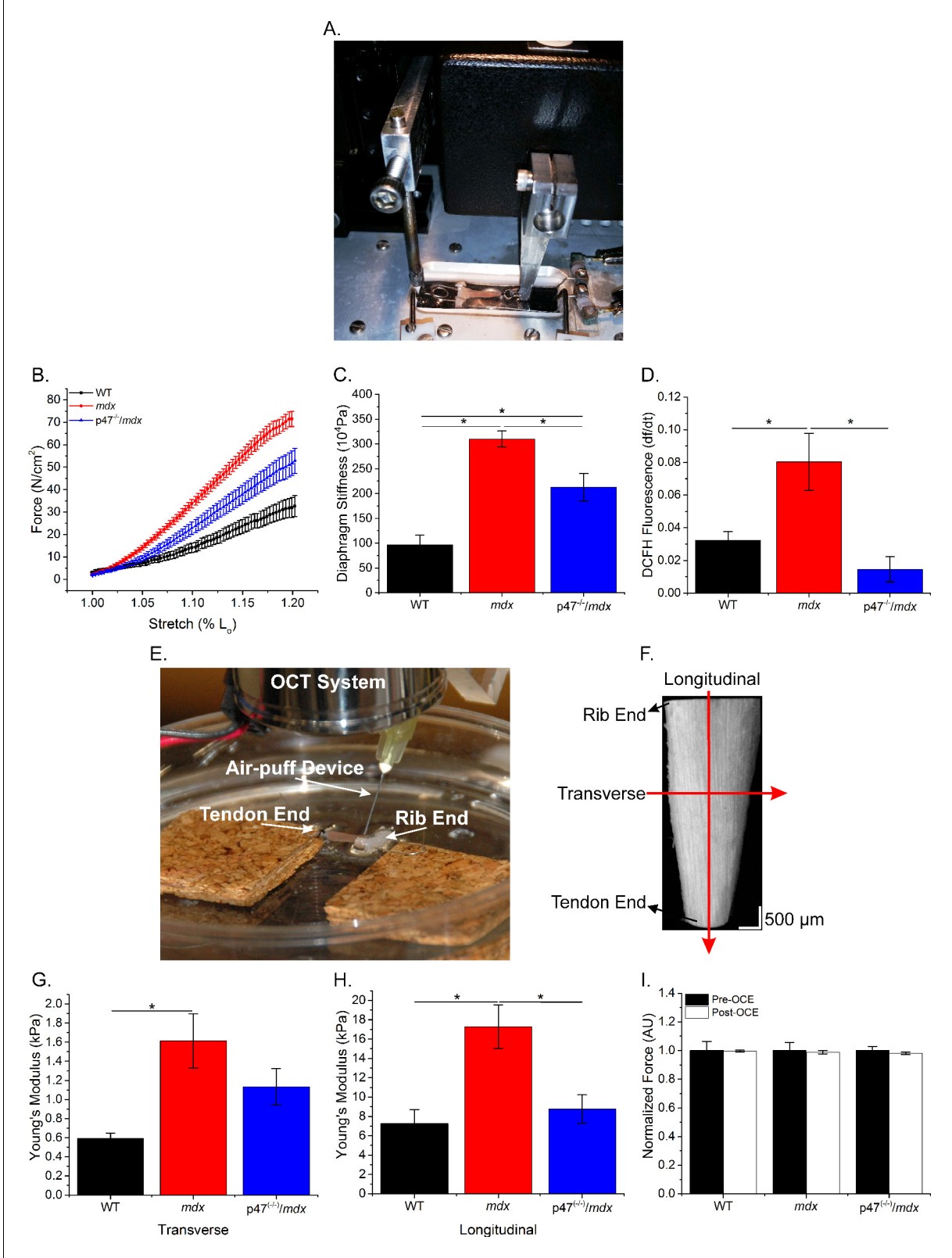

**Figure 3.** The lack of Nox2 ROS reduces muscle stiffness and stretch-induced ROS. (**A**) Image of the passive stretch experimental set-up. (**B**) Average passive diaphragm force recorded during stretch for each genotype. (**C**) Eliminating Nox2 ROS production reduced diaphragm tissue stiffness. (**D**) Stretch induced ROS in *mdx* muscle was elevated above WT levels and eliminated in p47$^{-/-}$/*mdx* diaphragm. (**E**) Image of the OCE experimental set-up. (**F**) Representative OCT image of the diaphragm taken prior to OCE experiments. (**G**) Transverse diaphragm muscle stiffness increased in *mdx*
*Figure 3 continued on next page*

Figure 3 continued

compared with WT mice; eliminating Nox2 ROS resulted in a decrease toward WT (p=0.09). (**H**) Genetic inhibition of Nox2 ROS reduced longitudinal diaphragm stiffness to WT values. (**I**) Muscle function was not altered following OCE measurements. $p \leq 0.05$ *Significant difference between groups in at least $n_{animals}$ = 6 per group.

The online version of this article includes the following video for figure 3:

**Figure 3—video 1.** Longitudinal.

https://elifesciences.org/articles/31732#fig3video1

addition, stiffness measured using OCE can detect changes in tissue elastic properties based on fiber orientation and indicate a direction-dependent response to alterations in tissue stiffness.

## Fibrosis is a major determinant of diaphragm stiffness

Increased DT-tubulin has been suggested to stabilize the microtubule network resulting in less dynamic microtubules thereby increasing tissue stiffness (*Kerr et al., 2015*; *Robison et al., 2016*). Our results demonstrate that while both α- and DT-tubulin are upregulated in dystrophic muscle the ratio of DT- to α-tubulin revealed no significant difference between groups (*Figure 1E*). A linear regression analysis demonstrated that fibrosis, DT-tubulin and α-tubulin significantly correlate to transverse and longitudinal diaphragm stiffness while the DT-/α-tubulin ratio only demonstrated a significant correlation with longitudinal stiffness (*Table 1*). A multiple linear regression analysis with either DT- or DT-/α-tubulin ratio and fibrosis revealed that the variance was no different than fibrosis alone (*Table 1*). Fibrosis accounted for 45% of the variance in the longitudinal and nearly 70% in the transverse direction. These data indicate that while tubulin content correlates with muscle stiffness, fibrosis accounts for the majority of the variance in muscle stiffness at the tissue level.

## Eliminating Nox2 ROS improves diaphragm muscle and respiratory function

Diaphragm muscle and respiratory function are compromised in *mdx* mice (*Huang et al., 2011*; *Ishizaki et al., 2008*; *Pal et al., 2014*; *Percival et al., 2012*). We previously have shown that eliminating Nox2 ROS production protected against diaphragm alterations in young (4–6 wks) *mdx* mice (*Pal et al., 2014*). Given muscle dysfunction in dystrophy is progressive, we wanted to determine whether eliminating Nox2 ROS provided protection against muscle/diaphragm dysfunction in older dystrophic mice. Here, we show that diaphragm function is impaired in adult (16–24 wks) *mdx* muscle and eliminating Nox2 ROS partially protected against the force deficits (*Figure 4A*). Eliminating Nox2 ROS in adult dystrophic muscle also protected against alterations in diaphragm fiber cross sectional area, fiber type and central nuclei *Figure 4—figure supplement 1*). These results, in combination with our previous data (*Pal et al., 2014*), indicate the lack of Nox2 ROS provides protection against pathophysiological alterations observed in dystrophic diaphragm muscle at different stages of disease pathology. In addition, eliminating Nox2 ROS protected against decrements in respiratory rate (f), minute ventilation (Mv), and peak inspiratory flow (PIF) in adult *mdx* mice (*Table 3*). A linear regression analysis demonstrated that fibrosis (*Figure 4B*) and both transverse and longitudinal diaphragm stiffness (*Figure 4—figure supplement 2*) significantly correlated with peak diaphragm force. A multiple linear regression analysis revealed when either transverse or longitudinal diaphragm stiffness was included with fibrosis, the variance was no different than fibrosis alone (*Table 2*). These data indicate Nox2-derived ROS drive alterations in *mdx* diaphragm which lead to diaphragm and respiratory dysfunction.

**Table 1.** Tubulin and stiffness correlations.

| Adj R² | Fibrosis | α-tubulin | β-tubulin | DT-tubulin | DT-/α-tubulin | MLR (fibrosis/DT) | MLR (fibrosis/ratio) |
|---|---|---|---|---|---|---|---|
| Transverse | 0.69 * | 0.46 * | 0.51 * | 0.51 * | 0.10 | 0.69 | 0.67 |
| Longitudinal | 0.44 * | 0.20 * | 0.40 * | 0.41 * | 0.19 * | 0.44 | 0.49 |

Most variables significantly correlated with both transverse and longitudinal stiffness. MLR revealed fibrosis accounted for the majority of the variance observed in either stiffness measure. $p \leq 0.05$ *Significant correlation in at least nanimals = 6.

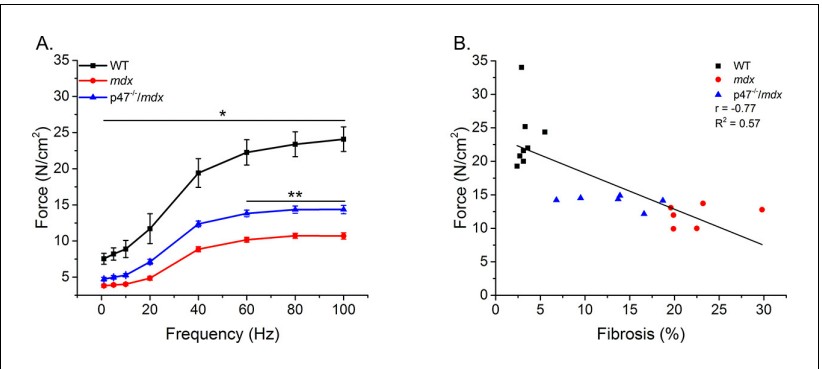

**Figure 4.** Eliminating Nox2 ROS protects against muscle and respiratory dysfunction. (**A**) WT was significantly different from *mdx* and p47[-/-]/*mdx* animals at all stimulation frequencies. The p47[-/-]/*mdx* animals were different from *mdx* at 60–100 Hz and trended toward significance at 40 Hz (p=0.098). (**B**) Fibrosis significantly correlated with muscle force. $p \le 0.05$ *Significant difference between groups in at least $n_{animals} = 6$.

The online version of this article includes the following figure supplement(s) for figure 4:

**Figure supplement 1.** Eliminating Nox2 ROS protects against phenotypic alterations in dystrophic diaphragm muscle.

**Figure supplement 2.** Linear correlation of stiffness measured by OCE and the peak force.

## Taxol-induced MT polymerization has no effect on tissue stiffness but induced ROS production

To further elucidate the role of the MT network in tissue stiffness and ROS production, we incubated WT diaphragm with Taxol to polymerize the MT network. We observed similar alterations in the MT network between Taxol-treated WT and *mdx* animals (*Figure 1F–I*; *Figure 5A–D*). Taxol increased MT density (*Figure 5D*) and resulted in disorganization of the MT network (*Figure 5B–C*). There was no difference in passive stiffness between Taxol and DMSO-treated diaphragm tissue (*Figure 5E–F*); however, there was a difference in stretch-induced ROS production (*Figure 5G*). These data, in combination with our previous data, support the idea that while alterations in the MT network increase ROS production, increases in DT-tubulin, MT density or MT disorganization do not influence tissue stiffness.

## Discussion

Froehner and colleagues (*Percival et al., 2007*) originally demonstrated MT disorganization in dystrophic muscle and its subsequent restoration with the re-introduction of mini-dystrophin. In *mdx* mice, the MT network becomes altered at approximately 7–8 wks of age (*Prins et al., 2009*) and remains altered with age (9–11 months) (*Kerr et al., 2015*). It has been suggested that alterations in the MT network lead to increased Nox2 ROS production and altered mechanotransduction in adult

Respiratory function.

|  | WT | *mdx* | p47[-/-]/*mdx* |
|---|---|---|---|
| f (breath/min) | 408.2 ± 14.5 * | 279.8 ± 18.3 | 377.3 ± 17.0 * |
| $T_v$ (ml) | 0.25 ± 0.009 | 0.24 ± 0.008 | 0.26 ± 0.012 |
| $M_v$ (ml) | 100.3 ± 5.6 * | 65.9 ± 17.6 | 99.2 ± 8.6 * |
| PIF (ml/s) | 7.6 ± 0.30 * | 5.9 ± 0.56 | 8.0 ± 0.51 * |
| PEF (ml/s) | 4.2 ± 0.25 | 3.2 ± 0.24 | 4.4 ± 0.39 * |
| $T_i$ (s) | 0.057 ± 0.002 * | 0.080 ± 0.007 | 0.057 ± 0.002 * |
| $T_e$ (s) | 0.129 ± 0.009 * | 0.190 ± 0.012 | 0.138 ± 0.008 * |

Dystrophic mice lacking Nox 2 ROS production maintained respiratory function similar to WT levels. $p \le 0.05$ *Significant difference vs. *mdx* in at least nanimals = 9.

**Table 2.** Force and stiffness correlations

| Adj $R^2$ | Fibrosis | MLR (fibrosis/trans) | MLR (fibrosis/long) | MLR (fibrosis/long/trans) |
|---|---|---|---|---|
| Force | 0.57 | 0.52 | 0.52 | 0.49 |

MLR revealed fibrosis accounted for a majority of the variance observed in diaphragm muscle function. $p \leq 0.05$ *Significant difference between groups in at least nanimals = 6.

$mdx$ muscle (**Kerr et al., 2015**; **Khairallah et al., 2012**). However, Nox2 ROS is upregulated prior to changes in the MT network (**Kerr et al., 2015**; **Prins et al., 2009**; **Whitehead et al., 2010**), raising the question whether increased Nox2 ROS drives changes in the MT network. In neurons, tubulin oxidation prevents MT polymerization (**Clark et al., 2014**; **Landino et al., 2004**; **Wilson and González-lez-Billault, 2015**); however, it is unclear what role increased ROS production plays in modulating the MT network of skeletal muscle. Our data show that diaphragm MT alterations are increased in adult dystrophic muscle and eliminating Nox2 ROS prevented the increase in α-, β-, and DT-tubulin content (**Figure 1B–D**), MT density (**Figure 1I**), MT disorganization (**Figure 1G–H**) and stiffness (**Figure 3C,G–H**) observed in $mdx$ mice. The MT network can be affected by muscle fiber type and regeneration (**Percival et al., 2007**; **Ralston et al., 1999**; **Ralston et al., 2001**); both of which are altered in dystrophic muscle. Here, we show that eliminating Nox2 ROS protected against

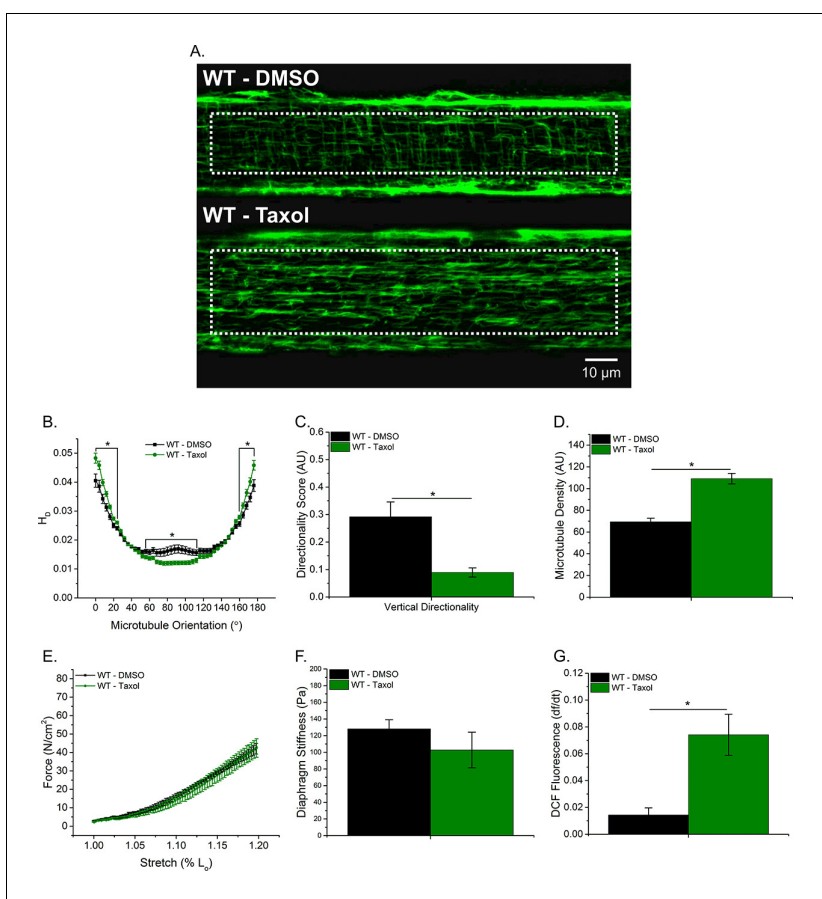

**Figure 5.** Taxol-induced MT polymerization has no effect on tissue stiffness but induced ROS production. (**A**) Representative images of MT network in control (DMSO) and Taxol-treated diaphragm (20 µM for 2 hr). (**B–D**) Taxol induced MT disorganization and increased microtubule density compared with control. (**E**) Average passive diaphragm force recorded during stretch was not affected by Taxol. (**F**) Polymerizing the MT network had no effect on diaphragm tissue stiffness. (**G**) MT network polymerization enhanced stretch-induced ROS in Taxol-treated diaphragm. $p \leq 0.05$ *Significant difference between groups in at least (**A–D**) $n_{animals}$ = 3 and $n_{fibers}$ = 15 and (**E–G**) $n_{animals}$ = 5.

alterations in fiber type switching and reduced central nuclei in dystrophic muscle. These data indicate Nox2 ROS, either directly or indirectly through alterations in fiber type or regeneration, is modulating the MT network.

Previous work has focused on either the cortical (*Percival et al., 2007*; *Prins et al., 2009* ) or some undetermined combination of the cortical and intermyofibrillar MT network (*Kerr et al., 2015*; *Khairallah et al., 2012* ). However, given the intermyofibrillar MT network surrounds the contractile apparatus, any alterations to this network likely affect force production. In addition, Nox2 is located in the plasma membrane and 60–90% of the plasma membrane in skeletal muscle is comprised by the t-tubules (*Eisenberg and Kuda, 1975*; *Mobley and Eisenberg, 1975*; *Peachey, 1965*). Therefore, the intermyofibrillar MT network may contribute more to muscle function and the mechanical activation of Nox2 ROS compared with the cortical MT network. To further explore whether the altered intermyofibrillar MT network influenced diaphragm stiffness and ROS production, we incubated WT diaphragm with Taxol. Polymerizing the MT network with Taxol resulted in increased intermyofibrillar MT density (*Figure 5D*) and disorganization (*Figure 5B–C*), similar to the diaphragm from *mdx* mice, but no change in tissue stiffness was detected. We found that Taxol increased stretch-dependent ROS production at the tissue level (*Figure 5F*); similar to what Khairallah et al. has shown in single FDB fibers (*Khairallah et al., 2012*). Taken together, we show that Nox2 ROS is an early event that modulates the MT network, potentially resulting in a feed forward mechanism where elevated Nox2 ROS production increases MT density and disorganization which in turn leads to additional Nox2 ROS production. We currently are investigating the mechanisms by which Nox2 ROS modulates the MT network.

Respiratory insufficiency in the DMD patient is caused by respiratory muscle weakness, leading to impaired ventilation through an inability to inhale and exhale fully, ultimately resulting in a need for mechanical ventilation. Dystrophic muscle is characterized by increased fibrosis and while some show no link between altered collagen and stiffness (*Chapman et al., 2015*; *Smith and Barton, 2014*) others have implicated fibrosis in decreased function and stiffness (*Cabrera et al., 2014*; *Desguerre et al., 2009*; *Ishizaki et al., 2008*; *Mead et al., 2014*; *Meyer and Lieber, 2011*; *Percival et al., 2012*; *Rowe et al., 2010*; *Wood et al., 2014*). Lateral force transmission through the endomysial layer of skeletal muscle has been shown to be important in overall force production (*Passerieux et al., 2007*; *Patel and Lieber, 1997*; *Purslow and Trotter, 1994*; *Trotter and Purslow, 1992*) and, in *mdx* mice, force is compromised in the transverse direction (*Kumar et al., 2004*; *Ramaswamy et al., 2011*). The endomysial layer also has increased levels of fibrosis which affects force production and correlates with the age of loss of ambulation in dystrophic muscle (*Desguerre et al., 2012*; *Desguerre et al., 2009*). Here, we show decreased diaphragm muscle (*Figure 4A*) and respiratory function (*Table 3*) and increased fibrosis (*Figure 2B*) and tissue stiffness (*Figure 3C,G–H*) in dystrophic muscle. Eliminating Nox2 ROS in dystrophic diaphragm muscle reduced fibrosis and tissue stiffness, increased force and prevented the decline in respiratory function. Highlighting the importance of cell-cell interactions, our data demonstrate a stronger correlation between force and transverse stiffness (*Figure 4—figure supplement 2*) and fibrosis and transverse stiffness than longitudinal stiffness (*Table 1*). These data indicate that fibrosis is a crucial factor altering tissue stiffness and force production resulting in impaired cell-cell interactions. Furthermore, a 26% increase in diaphragm force maintained respiratory function in the p47$^{-/-}$/*mdx* mouse, likely decreasing the need to place patients on a ventilator.

Several therapeutics designed to reduce fibrosis have proved beneficial in improving muscle function in dystrophic muscle (*Cabrera et al., 2014*; *Huebner et al., 2008*; *Percival et al., 2012*; *Turgeman et al., 2008*; *Whitehead et al., 2015*). Therefore, based on our data, it is conceivable that decreased fibrosis reduces transverse muscle stiffness, improving lateral force transmission and thereby overall muscle function. In addition, it has been suggested that fibrosis induces a feed forward loop causing collagen producing myogenic cells not to differentiate into terminal satellite cells; inhibiting myogenesis and enhancing fibrosis (*Alexakis et al., 2007*). These data are supported by the idea that progenitor cells take on a fibrogenic-like phenotype with aging; resulting in the loss of regenerative capacity in dystrophic muscle (*Biressi et al., 2014*; *Pessina et al., 2015*). The reduction in fibrosis observed by eliminating Nox2 ROS in dystrophic muscle may implicate a role for improved satellite cell activity given the reduced central nuclei and the increased CSA and Type 2B fibers (*Figure 4—figure supplement 1*) observed in the p47$^{-/-}$/*mdx* mice. In addition, we previously demonstrated eliminating Nox2 ROS improves autophagy in dystrophic muscle (*Pal et al., 2014*) and

autophagy is necessary for satellite cell differentiation and fusion (*Fortini et al., 2016*). Future experiments are needed to investigate the role of Nox2 ROS in the impairment of satellite cell function

Tissue stiffness in leg muscle mirrors changes in the MT network; becoming altered in *mdx* animals at approximately 7–8 weeks of age (*Wolff et al., 2006*) and remaining elevated in older animals (*Hakim et al., 2011*). Skeletal muscle stiffness has predominantly been assessed using atomic force microscopy (AFM) on single fibers (*Canato et al., 2010*; *Kerr et al., 2015*; *Mathur et al., 2001*; *van Zwieten et al., 2014*) or by passively lengthening muscle tissue (*Hakim and Duan, 2013*; *Hakim et al., 2011*; *Lopez et al., 2008*; *Rowe et al., 2010*). In C2C12 cells and isolated adult myofibers, alterations to the MT network increased cell stiffness, measured via AFM, and altered mechanotransduction (*Kerr et al., 2015*; *Khairallah et al., 2012*). However, AFM uses a point specific bending moment evaluating only the near-membrane mechanical properties at that point (*Kerr et al., 2015*). While this approach is vital for understanding intracellular contributions to single-cell signaling and near-membrane mechanics, it does not consider the ECM or cell-cell interactions in overall tissue mechanotransduction. Passive stretch takes into consideration both of these factors; however, it evaluates both the series (longitudinal) and parallel (transverse) elastic components together, making it difficult to assess the individual contributions to overall tissue stiffness. To address these limitations, we used two techniques to evaluate tissue stiffness, passive stretch and OCE. Interestingly, eliminating Nox2 ROS production partially prevented increases in tissue stiffness during passive lengthening (*Figure 3C*) similar to transverse stiffness measured using OCE (*Figure 3G*). In addition, we demonstrate a partial protection against force decrement (*Figure 4A*) and elevated transverse stiffness by eliminating Nox2 ROS production in the diaphragm (*Figure 3G*). These data highlight the importance of lateral (transverse) force transmission, and the significance of transverse stiffness in force production.

In isolated muscle cells, DT-tubulin, the de-tyrosinated form of α-tubulin, has been suggested to stabilize the MT network resulting in increased stiffness and reduced force (*Kerr et al., 2015*; *Robison et al., 2016*). However, MT formation is sensitive to alterations in the extracellular environment (*Myers et al., 2011*; *Putnam et al., 2003*; *Putnam et al., 2001*) implicating fibrosis in altering tissue stiffness. Previous work in neurons (*Bartolini et al., 2016*; *Cook et al., 1998*; *Infante et al., 2000*; *Khawaja et al., 1988*; *Morris et al., 2014*; *Skoufias and Wilson, 1998*; *Webster et al., 1990*) indicates DT-tubulin simply occurs temporally at the same time but was not the cause of MT stabilization. Furthermore, in skeletal muscle *Belanto et al. (2016)* have recently demonstrated that while DT-tubulin was elevated in *mdx* quadriceps muscle, the fraction of DT-/α-tubulin was no different than WT mice. Our data support the idea that while DT-tubulin is elevated in dystrophic diaphragm the DT-/α-tubulin ratio is no different (*Figure 1E*), indicating elevated DT-tubulin is a function of elevated α-tubulin and not the cause of stabilized MTs. Using the DT-/α-tubulin ratio as the indicator of stabilized MTs, our data demonstrate a significant but weak correlation with OCE longitudinal diaphragm stiffness and no correlation with transverse stiffness (*Table 1*). When included with fibrosis, while elevated DT-tubulin and the DT-/α-tubulin ratio correlated with tissue stiffness, MLR revealed neither influenced diaphragm tissue stiffness above fibrosis. These data suggest neither the absolute nor the relative amount of DT-tubulin influence tissue stiffness and fibrosis is the main determinant of diaphragm tissue stiffness.

Nox2 protein level and ROS production are upregulated early in dystrophic muscle prior to the inflammatory response (*Pal et al., 2014*; *Whitehead et al., 2010*). Previously, we have shown that Nox2 ROS production initiates a feed forward loop exacerbating Nox2 ROS production and inhibiting autophagic flux through activation of Src kinase (*Pal et al., 2014*). Interestingly, recent data by Froehner and colleagues (*Whitehead et al., 2015*) have shown that simvastatin reduced Nox2 protein levels, oxidative stress and fibrosis in *mdx* mice. Here we provide evidence for an additional feedforward mechanism where Nox2 ROS alters the MT network, which in turn exacerbates Nox2 ROS production. We also demonstrate that eliminating Nox2 ROS production alleviates many of the pathophysiological alterations, such as fibrosis, which occur in dystrophic diaphragm muscle. Taken together, there is compelling evidence that Nox2 ROS production is a central event in exacerbating disease pathology, implicating Nox2 as a viable therapeutic target in muscular dystrophy.

## Materials and methods

### Animals

C57Bl/6J (WT) and C57Bl/10ScSn-Dmd*mdx*/J (*mdx*) were purchased from Jackson Laboratories (Bar Harbor, ME) and bred following their breeding strategy. Mice lacking p47[phox] (B6(Cg)-Ncf1m1J/J, JaxMice) were crossed with *mdx* mice to generate *Ncf1*[-/-]::*mdx* (p47 [(-/-)]/*mdx*) mice (*Pal et al., 2014*)). At approximately 5 months of age and in accordance with National Institutes of Health guidelines and approved by the Institutional Animal Care and Use Committee of Baylor College of Medicine, mice were anesthetized by isoflurane (2%) inhalation and euthanized by rapid cervical dislocation followed by thoracotomy.

### Diaphragm passive stretch

Diaphragm muscle was surgically dissected and sectioned into diaphragm strips with the rib end attached to a fixed hook and the other to the lever arm of a dual-mode lever system (305C-LR-FP; Aurora Scientific Inc., Aurora, ON, Canada) using silk suture (4-0). The diaphragm was placed in a physiological saline solution containing (in mM): 2.0 $CaCl_2$, 120.0 NaCl, 4.0 KCl, 1.0 $MgSO_4$, 25.0 $NaHCO_3$, 1.0 $KH_2PO_4$, 10.0 glucose, pH 7.3 and continuously gassed with 95% $O_2$–5% $CO_2$ at 25° C. Muscle length was adjusted to elicit maximum twitch force (optimal length, $L_o$). A hand-held electronic caliper was used to measure $L_o$ and the lever arm was programmed to passively stretch the diaphragm strip to 120% of $L_o$ at 1 $L_o$/s for 5 min. At the end of the stretch protocol fiber bundles were removed from the rib, trimmed of excess connective tissue, blotted dry, and weighed. Muscle weight and $L_o$ were used to estimate absolute forces expressed as $N/cm^2$ (*Close, 1972*).

To determine tissue stiffness, the Veronda-Westman model (*Veronda and Westmann, 1970*) was employed to quantify Young's modulus for the first stretch. The Veronda-Westman model describes a nonlinear relationship between stress and strain and previously has been utilized to study the elasticity of a number of biological tissues, such as breast and skin (*Krouskop et al., 1998*; *Veronda and Westmann, 1970*). Assuming the diaphragm tissue as an incompressible Veronda-Westman material, under uniaxial tension, the axial stress σ is related to the resulted stretch λ through *equation 1*: (*Oberai et al., 2009*; *Pavan et al., 2010*)

$$\sigma = \frac{2E}{3}\left(\lambda^2 - \frac{1}{\lambda}\right)\left(e^{\gamma\left(\lambda^2 + \frac{2}{\lambda} - 3\right)} - \frac{1}{2\lambda}\right),$$ (1)

where λ = 1 + $\varepsilon$ ($\varepsilon$ is the strain), *E* is the Young's modulus of the diaphragm tissue at zero strain and γ is a nonlinear parameter representing the exponential increase rate of the Young's modulus over the increase of strain. Young's modulus was calculated through fitting the experimental data with *Equation 1* in Matlab (MathWorks; Natick, MA).

### ROS measurements

Diaphragm intracellular ROS was measured using 6-carboxy-2′,7′-dichlorodihydrofluorescein diacetate (DCFH-DA) (Invitrogen, Carlsbad, CA). Prior to stretch, the diaphragm was incubated with DCFH-DA for 30 min, washed using the physiological saline solution and de-esterified for an additional 30 min at 25°C. All cell-loading and imaging was performed in the dark to prevent light-induced oxidation of DCFH-DA. A Sutter Lamda DG-5 Ultra high-speed wavelength switcher was used to excite DCF at 470/20 nm and emission intensity was collected at 535/48 nm on a charge coupled device (CCD) Camera (CoolSNAP MYO, Photometrics, Tucson, AZ) attached to an Axio Observer (Zeiss) inverted microscope (20 × objective, 0.5 NA) at a rate of 0.2 Hz. Alterations in the rate of ROS production were baseline corrected and calculated over the final minute of the stretch period.

### Effect of taxol on tissue stiffness and ROS production

WT diaphragm tissue was incubated with 20 µM Taxol (Sigma-Aldrich, St. Louis, MO) or DMSO (Sigma-Aldrich, St. Louis, MO) control for 2 hr at RT. After 1 hr, the tissue was incubated with DCFH-DA, de-esterified and passively stretched as described above.

## Optical coherence elastography

Optical coherence elastography (OCE) is a novel technique for nondestructive assessment of mechanical properties of tissues (*Kennedy et al., 2017*; *Larin and Sampson, 2017*). The principle of OCE is based on producing a pressure wave on the sample and monitoring the propagation of the wave using phase-sensitive optical coherence tomography (OCT) imaging on nanometer scale. The velocity of the wave propagation in different directions along the surface is used to deduct tissue elasticity anisotropically (*Li et al., 2012*; *Wang et al., 2012*). A home-built OCE system was utilized which contains a focused air-puff device for tissue stimulation (*Wang et al., 2013*) and a spectral-domain OCT system to capture the tissue mechanical response (*Wang et al., 2014*). The air-puff system provided a highly localized (~150 μm in diameter), short-duration (~1 ms), and low-pressure (below 10 Pa) air stream to stimulate the surface of the diaphragm tissue in a noncontact fashion. The induced tissue displacement had a micro-scale amplitude. The OCT system had an axial resolution of ~5 μm in tissue, an imaging beam diameter of ~4 μm at the focal plane, and a displacement sensitivity of ~11 nm with the phase of the OCT complex signal. The tissue displacement over time was detected using the temporal phase profile from the OCT system. A previously reported shear wave imaging OCT approach (Wang and Larin) was utilized to capture the elastic wave propagation in a depth-resolved 2D field of view with a time resolution of 16 μs. Cross-correlation of tissue displacement profiles was used to measure the time delay formed by the wave propagation at different locations. The elastic wave velocity was thus quantified based on the slope from a linear fit of the time delay with respect to the wave propagation distance. A surface wave model (*Doyle, 1997*) that relates the sample Young's modulus $E$ to the wave velocity $C$ was utilized to estimate the tissue elasticity through *Equation 2*: (*Li et al., 2012*; *Wang et al., 2012*)

$$E = \frac{2\rho \times (1+\nu)^3 \times C^2}{(0.87 + 1.12\nu)^2} \tag{2}$$

where $\rho$ is the tissue density and $\nu$ is the Poisson's ratio; diaphragm density was 1060 kg/m$^3$ (*Mendez and Keys, 1960*). Due to the nearly incompressibility of soft tissue, the Poisson's ratio of 0.5 was utilized (*Mathur et al., 2001*). The averaged wave velocity value from 0 to 0.1 mm depth range from the tissue surface was used for calculation of the Young's modulus. For each diaphragm sample, the elastic wave assessment was conducted in the transverse and longitudinal directions of the muscle fiber.

## Ex vivo force measurements

Diaphragm muscle was surgically dissected from mice and sectioned into diaphragm strips with one end attached to a fixed hook and the other to a force transducer (F30, Harvard Apparatus) using silk suture (4-0) in a physiological saline solution continuously gassed with 95% $O_2$–5% $CO_2$ at 25°C. Diaphragm strips were incubated at 25°C for 10 min and optimal muscle length ($L_o$) and voltage ($V_{max}$) were adjusted to elicit maximum twitch force. Following a 5-min rest period, the diaphragm strip was stimulated at 150 Hz with pulse and train durations of 0.5 and 250 ms, respectively. Immediately after stimulation, $L_o$ was determined using a hand-held electronic caliper and the diaphragm strip was placed at $L_o$ in a 100 × 15 mm petri dish (VWR, Radnor, PA) for OCE measurements. Following OCE, the diaphragm was re-suspended from the force transducer at $L_o$ and after a 5-min rest period stimulated again at 150 Hz to ensure OCE measurements did not compromise the diaphragms functional properties.

To determine the force-frequency relationship, diaphragm strips were incubated at 30°C for 15 min and $L_o$ and $V_{max}$ were adjusted to elicit maximum twitch force. Following a 5-min rest period, force-frequency characteristics were measured at stimulation frequencies of 1, 5, 10, 20, 40, 60, 80, and 100 Hz every minute with pulse and train durations of 0.5 and 250 ms. At the end of the contractile protocol, $L_o$ was measured using a hand-held electronic caliper. Following both stimulation protocols, fiber bundles were trimmed of excess bone and connective tissue, blotted dry, and weighed. Muscle weight and $L_o$ were used to estimate cross-sectional area and absolute forces expressed as N/cm$^2$ (Close).

## Unrestrained whole-body plethysmography

Respiratory function was monitored in unrestrained mice using Buxco small animal whole-body plethysmography (Data Sciences International, New Brighton, MN) and FinePointe software (Data Sciences International, New Brighton, MN). The system was calibrated each day prior to data collection. On the day of data collection, animals were placed in individual chambers and given 30 min to acclimate; followed by 60 min of data collection. The software averaged the data over each minute and recorded a value every minute for 60 min. To ensure data was representative, breath frequency was used to ensure the mouse had not held its breath, buried its head under its body or was breathing too rapidly. Mean breath frequency was calculated and data which fell outside 1SD of the mean was excluded from the data analysis (*Roberts et al., 2015*).

## Western blot

Lysates from diaphragm tissue were extracted and quantified with the bicinchoninic acid (BCA) protein assay kit (Pierce, Rockford, IL), using BSA as the standard. Lysates were separated via SDS-PAGE and transferred to polyvinyldifluoride (PVDF) membranes. All tubulin blots were incubated in blocking buffer (5%, w/v, dried skimmed milk in Tris-buffered saline, pH 7.4, and 0.2% Tween 20; TBST) for 60 min and incubated overnight with anti-α-tubulin (Santa Cruz Biotechnologies), anti-β-tubulin (Cell Signaling Technology), anti-detyrosinatedtubulin (Millipore) and anti-GAPDH (Millipore) in blocking buffer. Fibronectin and collagen blots were blocked for 60 min in blocking buffer as above except with. 05% Tween 20 and incubated with anti-fibronectin (Millipore), anti-collagen (Abcam) and anti-GAPDH for 60 min at room temperature (RT). Tubulin and fibronectin blots were exposed to IRDye Secondary Antibodies (LI-COR Biosciences) diluted in TBST for 60 min at RT and washed again. The LI-COr Odyssey Infrared Imaging System was used for blot detection and ImageJ software for blot analysis. The collagen blot was probed with secondary antibodies; ECL anti-mouse IgG HRP (NA931, GE Healthcare) and ECL Anti-rabbit IgG HRP (NA93401, GE Healthcare) for 60 min at RT. The membrane was imaged using the Chemidoc touch with Clarity and Clarity Max ECL reagent (Bio-Rad, Hercules, CA). Image analysis was performed using Biorad Image Lab 6.0 software.

## Hydroxyproline assay

Diaphragm collagen content was measured using a hydroxyproline assay kit (Sigma-Aldrich, St. Louis, MO). Briefly, diaphragm tissue was homogenized and hydrolyzed in 200 µl of 6 M hydrochloric acid at 100°C for 3 hr. Hydrolysate was transferred to a 96-well plate (Corning, Corning, NY) and evaporated in an oven at 60°C. Following evaporation, the Chloromine T/Oxidation Buffer mixture was added to all wells and incubated for 5 min at RT. DMAB (4-(Dimethylamino) benzaldehyde) was diluted in a Perchloric Acid/Isopropanol solution, added to all wells, and incubated for 90 min at 60°C. A hydroxyproline standard curve (0–1.0 µg) was included in the assay to quantify hydroxyproline content in each sample. All samples, including the standard curve, were performed in duplicate and absorbance was measured at 560 nm. Results are reported as µg of hydroxyproline per mg of tissue (µg/mg).

## Immunofluorescence

For fiber-type, serial diaphragm sections of 12–14 µm thickness were sectioned at −24°C using a refrigerated cryostat (Shandon Cryotome E, Thermo). Sections were fixed with cold methanol for 20 min and incubated overnight in a humid box at 4°C with Anti-Type I (BA-F8) and anti-Type IIA (SC-71) antibodies purchased from Developmental Studies Hybridoma Bank (DSHB; Iowa City, IA). Sections were then incubated for 3 hr with IgG1 and IgG2b isotype-specific secondary antibodies (Invitrogen, Waltham, MA). Slides were mounted with VECTASHIELD anti-fade mounting media containing DAPI (Vector Laboratories, Berlingame, CA). Images were acquired using a CCD camera (Digital Sight DS-Fi1, Nikon) attached to an upright microscope (Nikon Eclipse 80i, 10 × objective, 0.45 NA). Images were analyzed using ImageJ software.

For α–tubulin staining, diaphragm tissue was fixed at $L_o$ using 10% neutral buffered formalin (VWR, Radnor, PA) for 2 hr at room temperature. The tissue was rinsed three times and stored in PBS (ThermoScientific, Waltham, MA) plus 1 mM EDTA (Invitrogen, Waltham, MA). Diaphragm fibers were mechanically dissociated from the fixed diaphragm strip into single fibers and placed in 35-mm

glass bottom culture dishes (MatTek, Ashland, MA) containing PBS plus 1 mM EDTA. Fibers were permeabilized with 0.1% Triton X-100 in PBS plus 1 mM EDTA for 10 min. After rinsing three times with PBS plus 1 mM EDTA, a blocking agent was added (0.1% saponin, 10% FBS in PBS plus 1 mM EDTA) for 1 hr at RT. Fibers were incubated with an Alexa-Fluor 488 conjugated α-tubulin antibody (Life Technologies, Waltham, MA) for 2 d at 4°C. Diaphragm fibers were washed with PBS and mounted with VECTASHIELD anti-fade mounting media containing DAPI (Vector Laboratories, Berlingame, CA) prior to microscopy. Fibers were imaged using a Zeiss LSM 780 confocal microscope (Zeiss, Oberkochen, Germany). Microtubule organization was analyzed using custom software (*Liu and Ralston, 2014*; software available through request to Dr Ralston) and microtubule density was assessed by summing 10 images from the intra-myofibrillar region of each fiber (>3 µm from surface), converted to a binary image and quantified using ImageJ software. Images were subjected to background subtraction and contrast enhancement using Image J for figure presentation only.

## Histology

Using a refrigerated cryostat (Shandon Cryotome E, Thermo), 12–14 µm thick serial sections were cut from the mid-belly region of the diaphragm at −24°C. Sections were stained using Masson's Trichrome for fibrosis and Hematoxylin and Eosin for cross-sectional area (CSA) and centralized nuclei. Images were acquired using a CCD camera (Digital Sight DS-Fi1, Nikon) attached to an upright microscope (Nikon Eclipse 80i, 10 × objective, 0.45 NA). Images were analyzed using ImageJ software.

## Statistical analysis

Data are reported as mean ±SEM, unless otherwise specified. A 1-way ANOVA was used to measure statistical differences between groups. A two-way RM ANOVA was used to determine statistical differences between groups for the force-frequency data. For CSA, a Kruskal-Wallis ANOVA was used to determine differences between groups. Tukey's post-hoc test was used when statistical differences were identified. Linear regression and multiple linear regression models were used to determine correlations between variables. Statistical analysis was performed in Origin Pro (OriginLab Corporation, Northhampton, MA) with significance set *a priori* at $p \leq 0.05$.

## Acknowledgements

The authors thank Drs. Wenhua Liu and Evelyn Ralston (National Institute of Arthritis and Musculoskeletal and Skin Diseases) for providing the directionality analysis program. Research reported in this publication was supported by the National Institute of Arthritis and Musculoskeletal and Skin Diseases of the National Institutes of Health under Award Number R01 AR061370 to GGR, the National Heart, Lung, and Blood Institute of the National Institutes of Health under Award Number R01 HL120140 to KVL and IVL and T32 HL007676 to JAL. Additional support was provided by the National Eye Institute of the National Institutes of Health under Award Number R01 EY022362 to KVL, the American Heart Association under Award Number 16POST30990070 to SW, and a Gillson Longenbaugh Foundation Award to GGR.

## Additional information

### Funding

| Funder | Grant reference number | Author |
| --- | --- | --- |
| National Institute of Arthritis and Musculoskeletal and Skin Diseases | AR061370 | George G Rodney |
| National Heart, Lung, and Blood Institute | HL007676 | James Anthony Loehr |
| National Eye Institute | EY022362 | Kirill V Larin |
| American Heart Association | 16POST30990070 | Shang Wang |
| Gillson Longenbaugh Founda- | | George G Rodney |

tion

| National Heart, Lung, and Blood Institute | HL120140 | Irina V Larina Kirill V Larin |
| --- | --- | --- |

The funders had no role in study design, data collection and interpretation, or the decision to submit the work for publication.

### Author contributions

James Anthony Loehr, Data curation, Formal analysis, Validation, Investigation, Methodology, Writing—original draft, Writing—review and editing; Shang Wang, Tanya R Cully, Rituraj Pal, Data curation, Formal analysis, Methodology, Writing—review and editing; Irina V Larina, Resources, Methodology, Writing—review and editing; Kirill V Larin, Resources, Supervision, Investigation, Methodology, Writing—review and editing; George G Rodney, Conceptualization, Resources, Supervision, Funding acquisition, Validation, Investigation, Methodology, Project administration, Writing—review and editing

### Author ORCIDs

James Anthony Loehr http://orcid.org/0000-0002-3524-5775
Shang Wang http://orcid.org/0000-0001-9447-719X
George G Rodney http://orcid.org/0000-0002-6968-1516

### Ethics

Animal experimentation: This study was performed in strict accordance with the recommendations in the Guide for the Care and Use of Laboratory Animals of the National Institutes of Health. All of the animals were handled according to approved institutional animal care and use committee (IACUC) protocols (#AN-5829) of Baylor College of Medicine.

### Decision letter and Author response

Decision letter https://doi.org/10.7554/eLife.31732.sa1
Author response https://doi.org/10.7554/eLife.31732.sa2

## Additional files

### Supplementary files

• Transparent reporting form

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
