## [Decision Letter]

Thank you for submitting your article "NADPH oxidase mediates microtubule alterations and diaphragm dysfunction in dystrophic mice" for consideration by *eLife*. Your article has been reviewed by three peer reviewers, one of whom, Stanley C Froehner (Reviewer #1), is a member of our Board of Reviewing Editors, and the evaluation has been overseen by Vivek Malhotra as the Senior Editor.

The reviewers have discussed the reviews with one another and the Reviewing Editor has drafted this decision to help you prepare a revised submission.

Summary

Work by this group and others suggested that increased Nox2 activity contributes to dystrophic muscle pathology and weakness in murine models of Duchenne muscular dystrophy (DMD). Increased Nox2 activity may result from changes in microtubule (MT) cytoskeleton organization, which may contribute to dystrophic muscle dysfunction including tissue stiffness and redox signaling imbalance. In this study, Loehr and coworkers explore the relationship between Nox2 activity, MT organization (particularly stability), and dystrophic skeletal muscle dysfunction (stiffness, weakness) and associated respiratory dysfunction using a previously published mouse model lacking both dystrophin and Nox2 expression. Loehr and coworkers ask if loss of Nox2 ROS could also alter the MT cytoskeleton or fibrosis to affect mdx diaphragm tissue stiffness, contractility and ultimately whole organism respiratory/ ventilatory function.

They show that loss of Nox2: (1) reduced disorganization and density of the MT cytoskeleton without impacting the fraction of glu-tubulin associated with more stable MTs and stiff tissue (2) qualitatively reduced fibrotic collagen deposition in diaphragm (3). decreased dystrophic muscle stiffness and stretch-induced Nox2 ROS synthesis indicated by conventional analysis of passive stretch properties and more novel elastography technique. (4) improved diaphragm muscle function (published previously) and improved indices of respiratory function including respiratory rate, minute ventilation and peak inspiratory flow. Tubulin content correlated with tissue stiffness, fibrosis accounts for most of variance in muscle stiffness and diaphragm specific force (across a range of stimulation frequencies), which is supported by many previous studies. (5) MT polymerization with the chemotherapeutic taxol induced ROS production (not surprisingly) but did not impact tissue stiffness strengthening the argument that MT stabilization is not the primary driver of tissue stiffness. Thus they provide evidence suggesting a feed forward loop where Nox2 activity may impact MT organisation, which in turn feeds back to increase Nox2 activity thereby promoting dystrophic skeletal muscle dysfunction.

Essential revisions

All reviewers agreed that the findings described in this manuscript are important and potentially appropriate for publication. However, all reviewers agreed that a major concern needs to be addressed with additional data.

The concern is the reliance on a single histological method – trichrome staining – to determine levels of fibrosis. Because of the firmly established patchy nature of fibrosis in mdx diaphragms, this method is not quantitative due to high susceptibility to sampling bias. Because many statistical analyses, and ultimately major conclusions of this study, depend on the accurate quantification of fibrosis, the use of this qualitative approach substantially undermines the conclusions. At least one other more quantitative method, such as western blotting or hydroxyproline measurements – preferably both – is needed to increase the confidence in this result. This shouldn't be too onerous to perform and feasible in the time allowed for resubmission.

Citations needed

The authors should cite the following studies.

a) The work of Whitehead et al. (PNAS 2015) showing that treatment of mdx mice with simvastatin reduces NOX2 levels, improves muscle function and reduces (and even reverses) fibrosis in the diaphragm.

b) If the authors used the directionality software provided by Liu and Ralston, they must have received a readme file that asked for an acknowledgement in potential publications. This is not out of pure vanity, but as an indication that their software was used, not a home-made version.

c) The first study to show aberrant MT organization in dystrophin null muscle fibers was PMID: 17714427 performed by Dr Froehner's lab. This study also showed that microdystrophin injection into 1 month old mdx mice is sufficient to prevent the subsequent disorganization of the MT cytoskeleton. Many groups subsequently confirmed these findings several years later.

1) In the paper that introduced the mdx/p47KO mouse (Pal et al., 2014) there were several characterizations. Its ROS production was impressively close to that of a wt animal but other properties were not (Ca influx seemed lower than either wt or mdx). Do the authors have evidence that ROS itself is the cause of microtubule and stiffness changes in mdx? Muscle microtubule organization is fiber type-dependent (Ralston et al. 1999 and 2001) and regeneration-dependent (Percival et al. 2007, Traffic 8, 1424). Since the mdx/p47KO muscles have more type IIx/IIb fibers than mdx muscles and fewer centrally nucleated fibers, these 2 factors may well contribute to the improvement of microtubules. In Pal et al., the data obtained with the mouse were supplemented with data obtained with inhibitors of Src or of p91. Are there microtubule changes with an acute treatment with p91 peptides?

2) Are the hindlimb muscle microtubules improved in the mdx/p47KO? Emphasis on the diaphragm, linked to mortality in DMD, is valuable. However, in Pompe Disease, a lysosomal storage disorder that affects skeletal and cardiac muscles, enzyme replacement therapy has improved cardiac function, preventing the death of children before age 2, but has not worked as well on skeletal muscles, leaving children and young adults with a longer but often miserable life.

3) The results showing that fibrosis rather than microtubules causes increased stiffness of the mdx diaphragm are compelling, and a warning that the whole muscle needs to be considered when modeling DMD. However the data also show that microtubules would contribute in experiments such as those of Khairallah et al. (2012) that use collagenase dissociated FDB fibers which presumably are freed of the fibrotic layer.

4) Most papers on the muscle microtubule network look at the dense layer of microtubules between the myofibrillar core and the sarcolemma (Percival et al., 2007; Prins et al., 2009; Khairallah et al., 2012 etc). Loehr et al., however, only examine the core microtubules. Given that both dystrophin and Nox2 are concentrated along the sarcolemma, why this choice? Are the results different (grid vs no-grid) if they look at the surface microtubules?

5) Clarification of overlap between the current study and what the authors have published previously. Using this model the authors previously reported (Pal et al., 2014, Loehr et al., 2016) that loss of Nox2 improved dystrophic muscle pathology (reduced central nucleation, macrophage infiltration, diaphragm weakness across a range of stimulation frequencies and normalized fiber type composition). Very similar data found in Pal et al., 2014 appears again in this study in Figure 4—figure supplement 2. The authors do state that"… eliminating Nox2 ROS production in young mdx mice can protect against diaphragm alterations (Pal et al., 2014)." But is it necessary to republish essentially the same findings?

6) In Figure 1F, are the mdx fibers centrally nucleated or not? Is there a quantitative difference in MT organization in Nox2 null mdx mice relative to mdx controls?

7) It is worth clarifying that the authors are mainly studying the subsarcolemmal MT cytoskeleton just under the, and not that found around the contractile machinery, which has a very different organization. This is important when interpreting the impact of taxol treatment, which will affect both MT pools.

8) Because of species-specific differences, and to the best of my knowledge, an absence of studies of MT or Nox2 in DMD patient muscle cells, the authors should be careful (e.g., cite appropriately) or avoid claiming (e.g. the first sentence of the Abstract) that what is phenotypically observed in mdx mouse models is recapitulated in DMD patients.

---

## [Author Response]

Essential revisionsAll reviewers agreed that the findings described in this manuscript are important and potentially appropriate for publication. However, all reviewers agreed that a major concern needs to be addressed with additional data.The concern is the reliance on a single histological method – trichrome staining – to determine levels of fibrosis. Because of the firmly established patchy nature of fibrosis in mdx diaphragms, this method is not quantitative due to high susceptibility to sampling bias. Because many statistical analyses, and ultimately major conclusions of this study, depend on the accurate quantification of fibrosis, the use of this qualitative approach substantially undermines the conclusions. At least one other more quantitative method, such as western blotting or hydroxyproline measurements – preferably both – is needed to increase the confidence in this result. This shouldn't be too onerous to perform and feasible in the time allowed for resubmission.

We agree with the concerns raised regarding trichrome staining. We have now added western blot data for Fibronectin and Collagen I along with measurements of hydroxyproline content to Figure 2. These new data confirm the trichrome data, showing that the increased fibrosis observed in the diaphragm from mdx mice is reduced upon Nox2 inhibition in the mdx/p47-/- mouse model.

Citations neededThe authors should cite the following studies.a) The work of Whitehead et al. (PNAS 2015) showing that treatment of mdx mice with simvastatin reduces NOX2 levels, improves muscle function and reduces (and even reverses) fibrosis in the diaphragm.

We have referenced and cited this work in the Discussion.

b) If the authors used the directionality software provided by Liu and Ralston, they must have received a readme file that asked for an acknowledgement in potential publications. This is not out of pure vanity, but as an indication that their software was used, not a home-made version.

We apologize for not appropriately acknowledging that we received and used the directionality algorithm provided by Dr. Ralston. In addition to the existing citation of that work, we have now added this to the acknowledgment section.

c) The first study to show aberrant MT organization in dystrophin null muscle fibers was PMID: 17714427 performed by Dr Froehner's lab. This study also showed that microdystrophin injection into 1 month old mdx mice is sufficient to prevent the subsequent disorganization of the MT cytoskeleton. Many groups subsequently confirmed these findings several years later.

We have now added the first report by Dr. Froehner’s group to our Discussion and citations.

1) In the paper that introduced the mdx/p47KO mouse (Pal et al., 2014) there were several characterizations. Its ROS production was impressively close to that of a wt animal but other properties were not (Ca influx seemed lower than either wt or mdx). Do the authors have evidence that ROS itself is the cause of microtubule and stiffness changes in mdx?

Previous reports have shown that the antioxidant NAC decreased some aspects of dystrophic pathology (Whitehead et al. J. Physiol. 2008), they did not access microtubule organization or cell/tissue stiffness. Here, we show with our p47-/-/mdx mouse, which lacks Nox2-dependent ROS production while still lacking dystrophin, that microtubule disorganization and tissue stiffness observed in the mdx mouse is likely due to ROS. We are currently testing whether treatment of mdx with antioxidants such as NAC might also provide protection against microtubule and stiffness changes.

Muscle microtubule organization is fiber type-dependent (Ralston et al. 1999 & 2001) and regeneration-dependent (Percival et al. 2007, Traffic 8, 1424). Since the mdx/p47KO muscles have more type IIx/IIb fibers than mdx muscles and fewer centrally nucleated fibers, these 2 factors may well contribute to the improvement of microtubules.

We agree with the reviewers and find this an interesting concept. Unfortunately we cannot delineate these contributions from the current data sets. We have acknowledged these possibilities/limitations in the Discussion.

In Pal et al., the data obtained with the mouse were supplemented with data obtained with inhibitors of Src or of p91. Are there microtubule changes with an acute treatment with p91 peptides?

This is very interesting question. Due to concerns on the bioavailability of the peptide throughout the depth of the tissue we decided not to pursue these experiments.

2) Are the hindlimb muscle microtubules improved in the mdx/p47KO? Emphasis on the diaphragm, linked to mortality in DMD, is valuable. However, in Pompe Disease, a lysosomal storage disorder that affects skeletal and cardiac muscles, enzyme replacement therapy has improved cardiac function, preventing the death of children before age 2, but has not worked as well on skeletal muscles, leaving children and young adults with a longer but often miserable life.

We agree with the reviewers that this is an important idea. This is an ongoing project by another student in the lab in which we are comparing multiple fiber types from limb muscles.

3) The results showing that fibrosis rather than microtubules causes increased stiffness of the mdx diaphragm are compelling, and a warning that the whole muscle needs to be considered when modeling DMD. However the data also show that microtubules would contribute in experiments such as those of Khairallah et al. (2012) that use collagenase dissociated FDB fibers which presumably are freed of the fibrotic layer.

We completely agree with the reviewers comments regarding the importance of the microtubules in cell signaling of isolated fibers. This concept was acknowledged in our discussion, which originally only mentioned C2C12 cells. We now include isolated single fibers (as done in the work by Khairallah et al., 2012) to this discussion.

4) Most papers on the muscle microtubule network look at the dense layer of microtubules between the myofibrillar core and the sarcolemma (Percival et al., 2007; Prins et al., 2009; Khairallah et al., 2012 etc). Loehr et al., however, only examine the core microtubules. Given that both dystrophin and Nox2 are concentrated along the sarcolemma, why this choice? Are the results different (grid vs no-grid) if they look at the surface microtubules?

Indeed, due to the idea that dystrophin binds the MT previous work has focused on either the cortical (Percival et al.; Prins et al.) or some undetermined combination of the cortical and intermyofibrillar MT network (Kerr et al.; Khairallah et al.). However, given the intermyofibrillar MT network surrounds the contractile apparatus, any alterations to this network likely affect force production. In addition, Nox2 is located in the plasma membrane and 60-90% of the plasma membrane in skeletal muscle is comprised by the t-tubules. Therefore, the intermyofibrillar MT network likely contributes more to muscle function and the mechanical activation of Nox2 ROS compared with the cortical MT network and therefore we focused on the intemyofibrillar MT network. We have added a discussion of this.

5) Clarification of overlap between the current study and what the authors have published previously. Using this model the authors previously reported (Pal et al., 2014, Loehr et al., 2016) that loss of Nox2 improved dystrophic muscle pathology (reduced central nucleation, macrophage infiltration, diaphragm weakness across a range of stimulation frequencies and normalized fiber type composition). Very similar data found in Pal et al., 2014 appears again in this study in Figure 4—figure supplement 2. The authors do state that"… eliminating Nox2 ROS production in young mdx mice can protect against diaphragm alterations (Pal et al., 2014)." But is it necessary to republish essentially the same findings?

Given muscle dysfunction in dystrophy is progressive, we wanted to determine whether eliminating Nox2 ROS provided protection against muscle/diaphragm

dysfunction in older dystrophic mice. This information is critical. If we found that eliminating Nox2 ROS was only protective in young and not adult we would have drawn a different conclusion regarding the therapeutic potential of targeting Nox2. Therefore, we do not feel this is “republishing essentially the same findings”. We have elaborated on this and hopefully the rationale is now clearer.

6) In Figure 1F, are the mdx fibers centrally nucleated or not? Is there a quantitative difference in MT organization in Nox2 null mdx mice relative to mdx controls?

While the mdx fibers have central nuclei, we decided to analyze density and organization in areas away from the nuclei to avoid confounding issues with the MT network around nuclei, which is also different around peripheral nuclei compared to the cortical MT network away from nuclei. Due to the complexities of generating Nox2 null mdx mice (both genes are X-linked) we elected to take the simpler approach of generating the p47-/-/mdx mouse line reported here.

7) It is worth clarifying that the authors are mainly studying the subsarcolemmal MT cytoskeleton just under the, and not that found around the contractile machinery, which has a very different organization. This is important when interpreting the impact of taxol treatment, which will affect both MT pools.

We are actually studying the MT network around the contractile machinery and not the cortical/subsarcolemmal MT network. We have clarified this and provided the rationale for this in response to point 4 above.

*8) Because of species-specific differences, and to the best of my knowledge, an absence of studies of MT or Nox2 in DMD patient muscle cells, the authors should be careful (e.g., cite appropriately) or avoid claiming (e.g. the first sentence of the Abstract) that what is phenotypically observed in mdx mouse models is recapitulated in DMD patients.*

We thank the reviewers for pointing out our overstatements. We have corrected this in the manuscript.